# Study on the Mechanism of the Adrenaline-Evoked Procoagulant Response in Human Platelets

**DOI:** 10.3390/ijms25052997

**Published:** 2024-03-05

**Authors:** Agata Gołaszewska, Tomasz Misztal, Adam Kazberuk, Tomasz Rusak

**Affiliations:** 1Department of General and Experimental Pathology, Medical University of Bialystok, Mickiewicza 2C, 15-230 Bialystok, Poland; 2Department of Physical Chemistry, Medical University of Bialystok, Mickiewicza 2A, 15-369 Bialystok, Poland; tomasz.misztal@umb.edu.pl (T.M.); tomasz.rusak@umb.edu.pl (T.R.); 3Department of Medicinal Chemistry, Medical University of Bialystok, Mickiewicza 2D, 15-959 Bialystok, Poland; kadam568@gmail.com

**Keywords:** adrenaline, ion exchangers, molecular mechanism, outside-in signaling, phosphatidylserine exposure, platelet, procoagulant response

## Abstract

Adrenaline has recently been found to trigger phosphatidylserine (PS) exposure on blood platelets, resulting in amplification of the coagulation process, but the mechanism is only fragmentarily established. Using a panel of platelet receptors’ antagonists and modulators of signaling pathways, we evaluated the importance of these in adrenaline-evoked PS exposure by flow cytometry. Calcium and sodium ion influx into platelet cytosol, after adrenaline treatment, was examined by fluorimetric measurements. We found a strong reduction in PS exposure after blocking of sodium and calcium ion influx via Na^+^/H^+^ exchanger (NHE) and Na^+^/Ca^2+^ exchanger (NCX), respectively. ADP receptor antagonists produced a moderate inhibitory effect. Substantial limitation of PS exposure was observed in the presence of GPIIb/IIIa antagonist, phosphoinositide-3 kinase (PI3-K) inhibitors, or prostaglandin E_1_, a cyclic adenosine monophosphate (cAMP)-elevating agent. We demonstrated that adrenaline may develop a procoagulant response in human platelets with the substantial role of ion exchangers (NHE and NCX), secreted ADP, GPIIb/IIIa-dependent outside-in signaling, and PI3-K. Inhibition of the above mechanisms and increasing cytosolic cAMP seem to be the most efficient procedures to control adrenaline-evoked PS exposure in human platelets.

## 1. Introduction

Numerous clinical observations indicate a positive correlation between stress and risk of thrombosis [1]. The stress reaction is connected with release of catecholamine hormone—adrenaline—into the bloodstream [2,3]. Adrenaline may modulate hemostatic response by changing the local hemodynamic conditions (through its vasoconstrictory properties) and by acting on platelets via G-protein-coupled α_2A_-adrenergic receptors (AR) [4,5]. Whilst many efforts have been taken to evaluate the potential of adrenaline to trigger platelet aggregation and thrombus formation [6,7,8,9,10], much less is known about engagement of the hormone in the development of the platelet procoagulant response and hence in the amplification of thrombin generation and fibrin deposition.

It is well documented that, within a thrombus, thrombin and fibrin are generated in a positive feedback loop, attributed to tenase (made of coagulation factors FVIIIa and FIXa) and prothrombinase (composed of factors FVa and FXa) complex formation within the phosphatidylserine (PS)-rich domains on the plasma membrane of activated platelets [11,12,13]. It is also well established that fibrin clots formed under high thrombin concentration are denser and more lysis-resistant compared to structures emerging in the presence of lower thrombin concentration, which may result in excessive fibrin deposition due to impaired fibrinolysis [14]. Recently, we reported that clinically relevant (low nanomolar) concentrations of adrenaline are capable of inducing PS exposure on the platelet plasma membrane and of notably enhancing platelet PS exposure evoked by subthreshold collagen, translating into amplified coagulation measured under static and flow conditions [15]. Increased platelet reactivity and upregulated thrombin generation are broadly recognized as major risk factors of thrombosis [16,17]. On the other hand, adrenaline has been proposed as a remedy for decreased platelet aggregability in patients receiving ticagrelor (antagonist of ADP receptor) [18,19]. Therefore, identification and characterization of procoagulant features of adrenaline are of vital cognitive and clinical value.

The activation of human platelets by adrenaline is mediated by α_2A_ adrenergic receptors, which are coupled with G_z_ protein, a subtype of G_i_ protein [20,21,22]. Canonically, adrenaline binding to α_2_–AR results in the inhibition of adenylate cyclase (AC) and therefore in a drop of cytosolic cyclic AMP (cAMP) level, which promotes activation of platelets by decreasing the activity of cAMP-dependent protein kinase (PKA), a major negative regulator of platelet reactiveness [5,23]. Other research also indicates the engagement of phosphoinositide-3 kinase (PI3-K), de novo synthesized TxA_2_, and Na^+^/H^+^ exchanger in adrenaline-induced platelet activation [7,24,25]. Thus, PS exposure in adrenaline-treated platelets is assumed to be more complex than limited to the “classical”, AC-related mechanism.

Consequently, the aim of this study was to investigate the molecular mechanism of the adrenaline-evoked procoagulant response in human platelets, exploring the confirmed and assumed signaling pathways related to adrenaline action.

## 2. Results

### 2.1. Inhibition of Only Adenylate Cyclase Is Not Sufficient to Produce Elevation of the PS-Exposing Platelets Number in the Presence of Subthreshold Collagen

As we have shown previously [15], adrenaline efficiently enhances PS exposure on human platelets exposed to subthreshold collagen. The most widely recognized effect of stimulation of the G_z_-coupled α-2-adrenergic receptor on human platelets is the inhibition of adenylate cyclase (AC) and therefore a decrease in cytosolic cyclic AMP (cAMP) level. Therefore, the first step in our investigation of such synergism between collagen GPVI and α-2-adrenergic receptors was to check whether substitution of adrenaline with a potent, selective, non-competitive AC inhibitor (SQ 22536, shown to be highly effective in the inhibition of platelet AC [26,27], Figure 1C) will produce a synergistic enhancement in PS exposure.

As shown in Figure 1—in contrast to the effect of adrenaline co-stimulation—selective inhibition of platelet AC by SQ 22536 was insufficient to induce a rise in the number of PS-positive platelets simultaneously exposed to subthreshold collagen. This suggests that synergistic elevation of PS exposure after simultaneous GPVI and α-2-adrenergic receptor stimulation is rather not associated with Gz-mediated inhibition of platelet AC but involves additional effector(s).

### 2.2. Adrenaline Produces Population of Procoagulant Human Platelets in a Concentration- and Time-Dependent Fashion

As shown in Figure 2, exposure to adrenaline (as a single agonist) results in the formation of P-selectin- and PS-exposing human platelets. Concomitant expression of these epitopes is commonly accepted as a characteristic feature of procoagulant platelets, in contrast to apoptotic platelets, which expose PS without P-selectin appearance (i.e., without platelet activation) [28].

To choose an effective adrenaline concentration producing strong PS exposure, we conducted experiments comprising the concentration and time-dependency. As shown in Figure 3A, exposure of platelets to adrenaline (0.001–10 μM, 45 min of exposure) resulted in a gradual increase in PS exposure, up to a concentration of 10 μM (producing 45 ± 11% of PS-exposing platelet in analyzed samples). Incubation of platelets with higher adrenaline did not elevate the number of PS-positive platelets, compared to 10 μM adrenaline (not shown). This effect was time-dependent, reaching its peak within 45 min (Figure 3B).

Consequently, to study the complex effect of adrenaline in the context of platelet PS exposure, in further experiments we decided to use a 10 μM adrenaline concentration (and 45 min incubation time), which is able per se to evoke high PS exposure, comparable to the effect exerted by a high concentration of a strong agonist (i.e., collagen, please compare Figure 1, Figure 2 and Figure 3) or by nanomolar adrenaline in combination with subthreshold collagen (please see [15,29]).

### 2.3. Effect of Selected Receptor Blockages on Adrenaline-Induced Platelet PS Exposure

To determine which platelet receptors are engaged in adrenaline-evoked PS exposure, we tested the effect of the antagonists of the following: α-2-adrenergic receptor (rauwolscine, phentolamine, BRL 44408), glycoprotein (GP)IIb/IIIa (tirofiban), P2Y_1_ receptor (MRS 2500), and P2Y_12_ receptor (PSB 0739). As shown in Figure 4, the most effective reduction in PS exposure was observed in the presence of rauwolscine and tirofiban, while the blockage of purinergic receptors produced a moderate inhibitory effect.

### 2.4. Impact of a Modulation of Different Signaling Pathways on Adrenaline-Induced PS Exposure

To study the significance of specific signaling pathways (related with platelet activation) in adrenaline-evoked PS exposure in platelets, we used a panel of inhibitors and modulators. As is shown in Figure 5, the most effective reduction (~90%) in PS exposure was observed in the presence of PGE_1_ (elevator of cytosolic cAMP level)—panel A. Strong inhibition of PS exposure (~60–80%) was recorded in the presence of calcium ion chelator (BAPTA-AM), an inhibitor of calcium efflux from a dense tubular system (2-APB)—panel B; inhibitor of Na^+^/H^+^ antiporter (EIPA), a blocker of the reverse mode of Na^+^/Ca^2+^ antiporter (KB-R7943)—panel B; inhibitors of PI3-K (wortmannin, LY 294002)—panel C. Low (~10–20%) to moderate (~30–40%) reduction in PS exposure was detected in the presence of RO 31-8220 (*pan* PKC inhibitor)—panel C. Inhibitors of thromboxane A_2_ synthesis (acetylsalicylic acid (ASA), indomethacin, panel D) or inhibitor of HCO_3_^−^/Cl^−^ exchanger (DIDS, panel B) did not decrease the number of PS-exposing platelets, after adrenaline treatment, significantly.

### 2.5. Strong Dependence of Adrenaline-Evoked PS Exposure on the Presence of Extracellular Calcium and Sodium Ions

Platelets suspended in Tyrode–Hepes buffer showed distinctly lower PS exposure following adrenaline treatment compared to the effect recorded in platelet-rich plasma (please compare Figure 3 with Figure 6A). In such an artificial environment, PS exposure upon adrenaline treatment was nearly completely abrogated by the absence of sodium or calcium ions (distinctly more deeply compared to collagen-stimulated platelets, Figure 6B) highlighting the importance of the influx of extracellular ions into platelet cytosol in this particular platelet response to adrenaline.

Indeed, the exposure of washed platelets to adrenaline produced a rise in cytosolic sodium ion concentration (measured indirectly as an alkalization of platelet cytosol due to NHE-attributed H^+^ removal), a phenomenon which was significantly reduced by the presence of α-2-adrenergic receptor inhibitor (rauwolscine) and likewise by the inhibition of NHE (by EIPA) and by using a GPIIb/IIIa receptor antagonist (tirofiban) (Figure 7A,B). Additionally, the exposure of washed platelets to adrenaline resulted in the increase in platelet volume (MPV), which was attenuated by the NHE inhibitor, EIPA (Appendix A). Similarly to sodium influx, the exposure of platelets to adrenaline resulted in a calcium ion influx into platelet cytosol, which was strongly reduced by rauwolscine, EIPA, inhibition of the reverse mode of NCX (by KB-R7943), and by tirofiban (Figure 7C,D).

## 3. Discussion

### 3.1. Role of AC and cAMP in Adrenaline-Evoked PS Exposure

Here, we present the results of our investigation into the mechanism of the adrenaline-evoked procoagulant response of human platelets and propose a model of this mechanism. We started with the question of whether a targeted inhibition of adenylate cyclase (AC)—the most widely recognized downstream effect of adrenaline α-2-adrenergic receptor stimulation—may result in significant enhancement of PS exposure on platelets exposed to collagen, in a similar fashion to adrenaline and collagen co-stimulation, which we reported previously [15]. To test such a possibility, we used SQ 22536, a cell-permeable AC inhibitor reported to efficiently decrease the cAMP level in resting human platelets [26], as well as in platelets exposed to PGE_1_ [30], or forskolin [30,31] (AC activators). However, we did not observe synergistic enhancement of PS exposure in the co-presence of SQ 22536 and collagen (Figure 1B). This leads to conclusion that adrenaline-induced enhancement of the platelet procoagulant response is not restricted to its canonical mechanism comprising inhibition of AC and involves additional effectors. On the other hand, pharmacological elevation of AC activity by PGE_1_ (mimicking physiological, prostacyclin-evoked, AC activation) resulted in a strong inhibition of adrenaline-evoked PS exposure (Figure 5A). This suggests that low AC activity (hence, low cAMP level) is a prerequisite, yet not the only element in the mechanism of adrenaline-triggered platelet PS exposure.

### 3.2. Receptors’ Engagement in Adrenaline-Triggered PS Exposure

After conducting a series of experiments including time and dose dependency between adrenaline presence and platelet PS exposure, we chose 45 min of incubation time and 10 µM adrenaline as optimal conditions to study the mechanism of adrenaline-induced PS exposure in human platelets (Figure 3). Such a high (supraphysiological) concentration of adrenaline was able to evoke strong PS exposure, virtually comparable to the effect exerted by collagen (at a relatively high concentration, i.e., 10 µg/mL), the most procoagulant one between physiological platelet agonists (please compare Figure 1, Figure 2 and Figure 3 from this paper and Figure 5A from another publication under the reference [32]). One important observation is that the mechanism of the adrenaline-evoked platelet procoagulant response relies on the activation of some platelet receptors, which seem to show supremacy over the others; blockage of the α-2 adrenergic receptor (AR) and GPIIb/IIIa (fibrinogen receptor) showed the strongest inhibitory effect on adrenaline-triggered PS exposure, while the antagonists of purinergic receptors (P2Y_1_ and P2Y_12_) exerted moderate inhibition of PS exposure (Figure 4). This emphasizes the central role of the direct interaction between adrenaline and platelet adrenergic receptors but also shows the non-redundant role of secreted ADP and the essential engagement of outside-in signaling, i.e., activation of additional signaling pathways (e.g., phosphoinositide-3 kinase (PI3-K)-dependent pathway), upon binding of fibrinogen to the activated GPIIb/IIIa receptors, which amplifies the initial stimulation. It has been reported that adrenaline-induced platelet aggregation and calcium signaling are amplified by ADP [24,33,34,35], but (to our best knowledge) such cooperation has never been studied in the context of procoagulant response.

Interestingly, in washed platelet samples, adrenaline—in contrast to collagen—produced distinctly lower PS exposure compared to the PRP condition (compare Figure 2 and Figure 6). One potential explanation is an insufficient amount of fibrinogen molecules in the washed platelets system (adrenaline is unable to trigger washed platelets aggregation [36] due to a lack of native or secreted fibrinogen), thus excluding outside-in signaling. Additional factors may include a partial platelet desensitization to the adrenaline action due to mechanical stress connected with centrifugation and washed platelets preparation.

### 3.3. Role of Ca^2+^ in the Development of the Procoagulant Response of Platelets Exposed to Adrenaline

Activation of platelets is coupled with the influx of ions into platelet cytosol [37,38,39,40]. Platelet activation requires a rise in intracellular Ca^2+^ concentrations, which results in morphological changes, secretion of the platelet granule’s contents, aggregation, and procoagulant response (PS exposure) [41]. The mechanism of physiological PS exposure in platelets is associated with the activity of scramblase (anoctamin-6, a product of the TMEM16F gene), a calcium-dependent transmembrane protein responsible for translocation of negatively charged PS from the inner to outer leaflet of the platelet plasma membrane [42,43,44,45,46]. Two separate sources of platelet-available calcium ions include intracellular storage (dense tubular system) and extracellular milieu (plasma). The first source is associated with Gq-protein-mediated activation of phospholipase C (PLC) and generation of the secondary messenger inositol triphosphate (IP_3_), which activates the IP_3_ receptor in membranes of the dense tubular system, resulting in liberation of Ca^2+^ into cytosol. Extracellular milieu, in turn, may serve as a source of ions due to the action of ion exchangers, with the particular role of Na^+^/H^+^ (NHE) and Na^+^/Ca^2+^ (NCX) exchangers.

The observation that suspending washed platelets in Ca^2+^-free medium or incubation of PRP with 2-APB (inhibitor of IP_3_ receptor in dense tubular system) strongly decreased adrenaline-triggered PS exposure suggests a vital role of calcium ions from both (extra- and intracellular) sources in the evoked platelet procoagulant response (Figure 5B and Figure 6A). Correspondingly, incubation of PRP with cell-permeable Ca^2+^ chelator, BAPTA-AM, strongly inhibited PS exposure (Figure 5). Adrenaline may activate platelet PLC indirectly by secreted ADP (which mildly activates PLC_β_ via the Gq-coupled P2Y_1_ receptor [47]) or by de novo synthesized thromboxane A_2_ (TxA_2_, released after platelet activation and stimulating PLC_β_ via Gq-coupled TP receptor [48]).

However, it seems that the TxA_2_-related pathway is unlikely in the presented model of platelet activation, since two inhibitors of TxA_2_ synthesis—ASA and indomethacin—did not decrease PS exposure in the adrenaline-treated platelets (Figure 5D). Conversely, the blockage of P2Y_1_ and P2Y_12_ receptors moderately decreased PS exposure (Figure 4). Since activation of P2Y_12_ has been linked with the prolongation of calcium signaling in thrombin-activated platelets (via PI3-K signaling [49]), the engagement of both purinergic receptors in adrenaline-induced PS exposure may be proposed. Another potential explanation could be that ADP activates additional pathways, next to PLC activation, including inhibition of AC and activation of PI3-K (both covered by adrenaline and ADP via appropriate Gα(i)- and Gβγ-associated events [50]), which might be necessary for a potent PS exposure upon adrenaline treatment. This is consistent with the observations of Storey et al. that the antagonists of purinergic receptors produce a significant decrease in adrenaline-triggered platelet aggregation [51]. Also, in our previous report, we showed that blockage of platelet secretion (by HAuCl_4_) eliminated the second wave of adrenaline-induced platelet aggregation, indicating the important role of secreted ADP in this response [52].

Collectively, this suggests that adrenaline-evoked PS exposure may depend upon secretion of a secondary agonist—ADP but not TxA2—that further stimulates PLC via Gq-coupled receptors (P2Y_1_) and inhibits AC via Gi-coupled receptors (P2Y_12_).

### 3.4. Significance of Na^+^ and Platelet Swelling in Adrenaline-Produced PS Exposure

The engagement of extracellular calcium ions in adrenaline-evoked PS exposure was measured in relation to sodium ions since activation of platelets by principal agonists is associated with sodium ion influx into platelet cytosol [53,54]. According to Sweatt et al., exposure of platelets to adrenaline results in activation of type 1 Na+/H+ exchanger (NHE1), leading to sodium accumulation and alkalization of platelet cytosol due to concomitant H^+^ removal [55]. Literature data suggest that NHE activation and influx of sodium ions are correlated with platelet PS exposure [56,57]. One may therefore suppose that NHE also plays a vital role in the case of adrenaline-evoked platelet PS exposure. Indeed, incubation of platelets with EIPA (NHE inhibitor) resulted in a strong inhibition of adrenaline-induced PS exposure (Figure 5B). Correspondingly, in contrast to collagen, we recorded a lack of adrenaline-triggered PS exposure in washed platelets suspended in a sodium-free medium, i.e., in a condition where NHE activity was arrested (Figure 6A,B). Overloading of platelet cytosol with sodium ions is a prerequisite to activation of the reverse mode of the Na^+^/Ca^2+^ exchanger (NCX). In the reverse mode, NCX removes sodium ions from the cytosol, exchanging them for extracellular calcium ions in the following manner: 3Na^+^ → Ca^2+^ [58]. Both ions (sodium and calcium) have been recognized as crucial factors during platelet swelling, ballooning (forming of a balloon-shaped morphology of particularly strongly procoagulant platelets), and PS exposure in platelets adhered to collagen [42]. NHE plays a pivotal role in controlling platelet swelling—a phenomenon correlating with platelet ballooning and PS exposure [29,42,56,57,59,60]—by generation of an osmotic gradient, driving water molecules into platelet cytosol via aquaporins (AQPs). Previously, we reported enhanced ballooning of platelets adhered to collagen after preincubation with adrenaline [15]. Here, we demonstrate that adrenaline per se is able to evoke gradual platelet swelling (measured as a time-dependent increase in mean platelet volume), which was inhibited by EIPA (Appendix A). This indicates that platelet swelling, and the plasma membrane extension associated with it, facilitates PS exposure not only in (previously reported) the collagen- and hyperglycemia-induced platelet procoagulant response [29,42] but also in the presence of adrenaline.

Since startup of the reverse mode of NCX is a secondary event to sodium influx (via NHE), inhibition of NHE should attenuate not only sodium but also calcium influx into platelet cytosol (assuming that the reverse mode of NCX is vital in adrenaline-induced calcium entry) and, as a result, PS exposure. Measurements of PS exposure but also sodium and calcium entry into the cytosol of adrenaline-treated platelets—in the presence of NHE or NCX inhibitors—confirmed such a possibility (please compare Figure 4B and Figure 7A–D). Additionally, the inhibition of sodium and calcium influx into adrenaline-exposed platelets—and the abolition of PS exposure corresponding with it—was observed not only as a result of adrenergic receptors blockage or NHE/NCX inhibition but also in the presence of the GPIIb/IIIa antagonist (Figure 4 and Figure 7A–D). These results indicate that adrenaline-evoked activation (leading to PS exposure) is more strongly connected with the ionic composition of extracellular milieu and with ion influx into platelet cytosol than in the case of collagen-triggered PS exposure. In comparison to adrenaline, collagen-evoked PS exposure was moderately reduced by the absence of extracellular sodium ions, which is in line with results obtained by Roberts et al., where the authors examined the requirements for extracellular ions in the context of collagen-induced platelet aggregation [39].

### 3.5. Role of GPIIb/IIIa-Mediated Outside-in Signaling in Adrenaline-Triggered PS Exposure

A connection between GPIIb/IIIa and NHE/NCX has been proposed by Banga et al. in the context of the initial events during platelet activation by adrenaline, i.e., phospholipase A_2_ activation, thromboxane synthesis, and secondary (via TxA_2_ receptor) phospholipase C activation [48]. NCX, in turn, was investigated in the context of GPIIb/IIIa activation, a prerequisite for platelet aggregation and outside-in signaling. Studies by Shiraga et al. revealed that inhibition of the reverse mode of NCX distinctly reduced GPIIb/IIIa activation and thus platelet aggregation [61,62]. The results presented here also suggest the possibility of the existence of an inverse relationship, i.e., blockage of GPIIb/IIIa by tirofiban might strongly reduce adrenaline-evoked sodium and calcium influx into platelet cytosol via NHE and NCX (in the reverse mode), respectively. Hence, we propose a scenario where binding adrenaline to α_2_-AR evokes rapid activation of a relatively moderate number of GPIIb/IIIa (covering the first phase in the characteristic biphasic aggregation of adrenaline-exposed platelets [6]), which induces signaling pathways associated with outside-in signaling, resulting in the stimulation of NHE activity and further resulting in the startup of a reverse mode of NCX. As a result of the above events, sodium and calcium ions emerge into platelet cytosol, triggering additional pathways manifesting into full activation of GPIIb/IIIa and the second (irreversible) phase of aggregation or the procoagulant response (PS exposure).

The strikingly effective inhibition of adrenaline-evoked PS exposure by the GPIIb/IIIa antagonist deserves a comment. The classical context of GPIIb/IIIa antagonism is platelet aggregation, and the aspect of platelet procoagulant response modulation by a GPIIb/IIIa-dependent signaling is less commonly recognized. During platelet activation in a prothrombotic environment (e.g., collagen-rich surface), two populations of activated platelets—proaggregatory and procoagulant—are formed [12]. Procoagulant (PS-exposed) platelets are characterized by initial activation of GPIIb/IIIa, which subsequently become inactivated [63,64,65]. The initial activation of GPIIb/IIIa seems therefore to be an indispensable event in developing both proaggregatory and procoagulant phenotypes. Experimental evidence supporting the idea that GPIIb/IIIa-related outside-in signaling might be a vital element in the procoagulant response has gained prominence. According to Topalov et al., two subtypes of procoagulant platelets (with high or low sustained calcium level) are formed upon physiological activation, which are controlled by GPIIb/IIIa [66]. It is noteworthy that, referring to Ilveskero et al., platelets adhered to fibrin(ogen) clots exert strong procoagulant activity—indicating the procoagulant nature of platelet–fibrin(ogen) interactions—which could be controlled by GPIIb/IIIa blockage [67,68]. More accurately, GPIIb/IIIa antagonists, i.e., tirofiban, eptifibatide, and abciximab, were shown to distinctly impair not only platelet aggregation (primary effect of these drugs) but also the procoagulant response (PS exposure and platelet-related thrombin generation) in well-established experimental models [69,70,71,72]. Based on the body of literature and the results presented here, we therefore postulate that GPIIb/IIIa-mediated signaling is crucial to the development of platelet PS exposure triggered by, at least some, physiological stimuli, including adrenaline.

Interestingly, inhibition of the HCO_3_^−^/Cl^−^ exchanger by DIDS, which was reported as an inhibitor of the second phase of adrenaline-evoked platelet aggregation by Spalding et al. [73], did not modulate adrenaline-evoked PS exposure (Figure 5B). This is similar to the (lack of) effect of TxA_2_ synthesis blockage. One may speculate that the initial activation of a part of GPIIb/IIIa receptors upon adrenaline treatment (correlating with the first phase of aggregation and providing outside-in signaling) meets the condition where PS exposure can occur.

### 3.6. Non-Redundant Role of PI3-K in Adrenaline-Evoked PS Exposure

PS exposure in adrenaline-treated platelets was strongly inhibited by two structurally unrelated phosphoinositide-3 kinase (PI3-K) inhibitors—wortmannin and LY 294002 (Figure 5C). PI3-K is a group of lipid kinases, which phosphorylate the 3′ position of the hydroxyl group in the inositol group of the phosphatidylinositol (PI), resulting in 3-phosphoinositides, a group of secondary messengers critical to various aspects of platelet biology as a part of the PI3-K/Akt signaling pathway [74]. Human platelets express at least seven [74,75,76] isoforms of PI3-K belonging to classes I, II, and III, among which the class I PI3-K α and β isoforms play a key role in thrombus formation on collagen-coated surfaces under flow and in PS exposure [77], while β and γ isoforms have been shown to play pivotal roles in the P2Y_12_-dependent activation of GPIIb/IIIa and thus in stable platelet aggregation [78,79]. Adrenaline has been reported to enhance PI3-K activity in platelets stimulated by thrombin or by the thrombin receptor activating peptides [24]. Keeping in mind that activation of PI3-K is a common pathway attributed to α_2_-AR and P2Y_12_ receptors [80,81,82] but also for GPIIb/IIIa-dependent outside-in signaling [83,84,85,86,87,88,89], the hypothesis might be proposed that a strong reduction of adrenaline-evoked PS exposure by tirofiban is, at least partially, associated with a decrease in total PI3-K activity.

It is noteworthy that numerous efforts to define the role of PI3-K isoforms in thrombus formation have been made and potential benefits of specific PI3-K inhibitors are being discussed [76,90]. Our results support the hypothesis that PI3-K might be considered as a promising target for future antiplatelet drugs, exerting both antiaggregatory and antiprocoagulatory effects.

### 3.7. How Does This Study Adhere to Clinical Reality?

The micromolar concentration of adrenaline used in this study does not realistically appear in the bloodstream in vivo. However, when used in a supraphysiological concentration, adrenaline is able to trigger strong platelet activation, including procoagulant (not apoptotic) PS exposure. We claim that this legitimizes using such high concentrations of adrenaline as an examination tool to study Gi(_z_)-related and unrelated signaling pathways. Considering the prolonged time (up to 45 min) required to develop a full response upon adrenaline exposure (observe at 10 μM, as well as at 1 μM adrenaline), it seems to be possible that the time of platelet sensitization is also prolonged. This, in turn, might be vital while facing a subthreshold concentration of a physiological agonist, even some time after direct interaction with adrenaline. The potential impact of monoamine oxidase inhibitors (which are likely to increase adrenaline concentration in the blood) might be an important direction of future studies.

Previously, we reported that the incubation of platelets with clinically relevant adrenaline (1–10 nM) and a subthreshold concentration of collagen resulted in strong, synergistic PS exposure, which can be counteracted by agents representing the most commonly used antiplatelet strategies, i.e., cyclooxygenase inhibitor, GPIIb/IIIa blocker, and P2Y_12_ antagonist [15]. Hereby, the presented results indicate that a supraphysiological concentration of adrenaline may evoke strong platelet PS exposure unable to be efficiently controlled by TxA_2_ synthesis blockers and the P2Y_12_ antagonist, i.e., agents used in single-drug and in dual antiplatelet pharmacotherapy. Is such a situation possible in vivo? It has been proposed that adrenaline supplementation may be useful for reversing the antiaggregatory effect of ticagrelor (P2Y_12_ antagonist) by triggering signaling pathways which are analogous to the unavailable P2Y_12_, i.e., inhibition of AC and activation of PI3-K. Efficient concentrations of adrenaline, restoring platelet aggregability in response to ADP, were estimated to be ~770 nM in vitro and ~20 nM in a model of controlled adrenaline infusion into volunteers’ bloodstreams for ex vivo examinations [18,19,50]. However, we would like to accentuate that a lack of clinical procedures to evaluate patient-specific sensitivity to adrenaline, in the context of platelet PS exposure and prothrombotic conditions associated with it, makes this consideration limited. Therefore, it might be a dilemma to evaluate what concentration of adrenaline is still beneficial and distant from the “supraphysiological” effect, where procoagulant conditions might appear. Based on the presented here results, such threat could be–potentially–only partially controllable by TxA_2_ synthesis inhibitors or P2Y_12_ antagonists. An alternative candidate to restoring platelet aggregability in ticagrelor-receiving patients could be bentracimab (a phase 3 clinical trial monoclonal antibody specific to ticagrelor); however, its usage is likely to be limited due to availability and resource-related issues [91].

## 4. Materials and Methods

### 4.1. Chemicals

Hepes, apyrase, adrenaline [(−)-epinephrine (+)-bitartrate salt)], acetylsalicylic acid (ASA), bovine serum albumin (BSA), human fibrinogen, PGE1, LY 294002, wortmannin, digitonin, nigericin, EIPA, BCECF-AM, acetazolamide, indometacine, phentolamine, TRIS (tris(hydroxymethyl)aminomethane), DIDS, Ponceau Red, and TBST buffer—Millipore, Tirofiban (Aggrastat) were acquired from Sigma (Merck KGAA, Darmstadt, Germany). Type I collagen was acquired from Chrono-log. Phycoerythrin (PE)-antihuman CD41a antibody, FITC-antihuman CD62P (P-selectin), and FITC-annexin V were acquired from Becton Dickinson (Franklin Lakes, NJ, United States). Rauwolscine hydrochloride, PSB 0739, KB-R7943 mesylate, 2-APB, MRS 2500, SQ 22536, and BRL 44408 maleate were acquired from Tocris. BAPTA-AM was acquired from Abcam (Cambridge, UK). Fura-2-AM and AlexaFluor(AF)647-annexin V were acquired from Thermo Fisher Scientific (Waltham, MA, United States). Bovine thrombin was acquired from was acquired from Synthaverse S.A., Lublin, Poland. Other chemicals were acquired from Sigma (Merck).

### 4.2. Blood Collection and Preparation

Venous blood was collected from healthy volunteers with minimum trauma and stasis via a 21-gauge needle (0.8 × 40 mm) into 10 mL polypropylene tubes containing 1 mL of 130 mM trisodium citrate. All procedures were conducted in accordance with the principles of the Declaration of Helsinki, and this study was approved by the local ethics committee on human research (R-I-002/430/2019). Platelet-rich plasma (PRP) was obtained by centrifugation of whole blood at 200× *g* for 20 min. Preparation of washed platelets was performed as described in [28]. Sodium-free Tyrode-HEPES buffer was prepared by substituting Na^+^ with equimolar N-methyl-D-glucamine to maintain osmolality.

### 4.3. PS and P-Selectin Exposure Quantification by Flow Cytometry

In platelet phosphatidylserine (PS) exposure determination experiments, we used PRP due to the fact that development of a procoagulant platelet population in vivo takes place at close proximity to vessel wall, i.e., in a platelet-rich and erythrocyte-free zone [92]. PRP samples were incubated without any additions (control), with adrenaline (10 μM or 0.001–10 μM range, in dose-dependency experiments), or with a range of inhibitors/antagonists followed by the addition of adrenaline (10 μM fixed conc.). After 30 min of incubation (5–60 min in time-dependent experiments) at room temperature, PS exposure was assessed by flow cytometry, as described in detail previously [28]. In some experiments, for comparative reasons, collagen was used as an agonist at a concentration of 250 ng/mL (“low collagen”) or 10 μg/mL (“high collagen”). In additional experiments, washed platelet samples were preincubated with adrenaline (10 μM for 45 min) or with collagen (10 μg/mL for 15 min) in an environment (Tyrode buffer) without Na^+^ or Ca^2+^ ions before cytometric determination of PS exposure. Concomitant detection of PS exposure and P-selectin was performed in PRP.

### 4.4. Fluorimetric Determination of Na^+^/H^+^ Exchange and Calcium Rise in Platelets Exposed to Adrenaline

For evaluation of N^+^/H^+^ exchanger (NHE) activity, we monitored changes in the cytosolic pH of BCECF-loaded platelets exposed to adrenaline [93]. The platelets were loaded with BCECF acetoxymethyl ester (10 μM final conc.) mixed with Pluronic 137 (0.005% final conc.) for 30 min at 37 °C. Then, 1 μM prostaglandin E_1_ was added, and washed platelets were separated from PRP as described above. Monochromators were set at 495 nm for the excitation wavelength and at 530 nm for the emission wavelength. Washed platelets were stimulated in a thermostated (37 °C) fluorimeter cuvette with stirring (800 rpm) using a Hitachi F-7000 spectrofluorometer (Hitachi Corp., Tokyo, Japan). Stimulators were added directly to the stirred platelet suspension through a port in a fluorimeter cover by using a microsyringe. To measure calcium ion influx into platelet cytosol, PRP was incubated with 2 μM (final conc.) Fura-2/AM and Pluronic-137 (0.005% final conc.) for 45 min at room temperature in the dark with gentle agitation. After that time, washed platelets were prepared. Measurement of Fura-2 fluorescence in stirred platelet suspension (in the presence of 1 mM Ca^2+^ in reaction buffer) was performed at 37 °C using the following monochromator settings: 339 nm for excitation and 500 nm for emission. The Fura-2 responses were calibrated to obtain [Ca^2+^]cyt as described by Siffert et al. [93].

### 4.5. Measurement of Platelet Aggregation

Platelet aggregation in PRP was measured optically as an increase in light transmittance associated with aggregation progress [94] using Elvi Logos 840 (Elvi S.p.a., Milano, Italy) aggregometer connected to a personal computer via e-corder 401 (eDAQ Pty Ltd., Denistone East, Australia). Aggregation was monitored using eDAQ Chart software (Denistone East, Australia)—version 5.2.12 (released on 28 August 2006).

### 4.6. Electronic Measurements of Platelet Volume

Aliquots of standardized washed platelet suspension (2 × 10^8^ cells/mL) supplemented with adrenaline (1 µM), adrenaline + EIPA (100 µM), or without any addition (control) were incubated up to 40 min at 37 °C. At selected time points, platelets were fixed with paraformaldehyde (2% final conc.), and their mean volume (MPV) was measured by using a hematologic analyzer (Coulter Electronic GmbH, Krefeld, Germany).

### 4.7. Data Analyses

Data were evaluated using GraphPad Prism 5 (GraphPad Software, version 5.03). Differences between the two groups were assessed by Mann–Whitney U test. The data are shown as the median (1st–3rd quartile) of the number of determinations (n) or as a percentage compared to the control. In all experiments, a *p* value of <0.05 was considered to be significant.

## 5. Conclusions

Collectively, the presented results hereby suggest the existence of a mechanistic association between the binding of adrenaline to platelet α_2_-AR, activation of GPIIb/IIIa (resulting in outside-in signaling, including PI3-K activation), chronological activation of NHE and the reverse mode of NCX, and, ultimately, PS exposure. In this model, secreted ADP enhances the initial signal by further AC inhibition and PI3-K activation, which altogether produce strong PS exposure in human platelets treated with (supraphysiological) adrenaline. Inhibition of the above elements and increasing cytosolic cAMP are the most efficient strategies to reduce adrenaline-evoked platelet procoagulant (summarized in Figure 8).

## Figures and Tables

**Figure 1 ijms-25-02997-f001:**
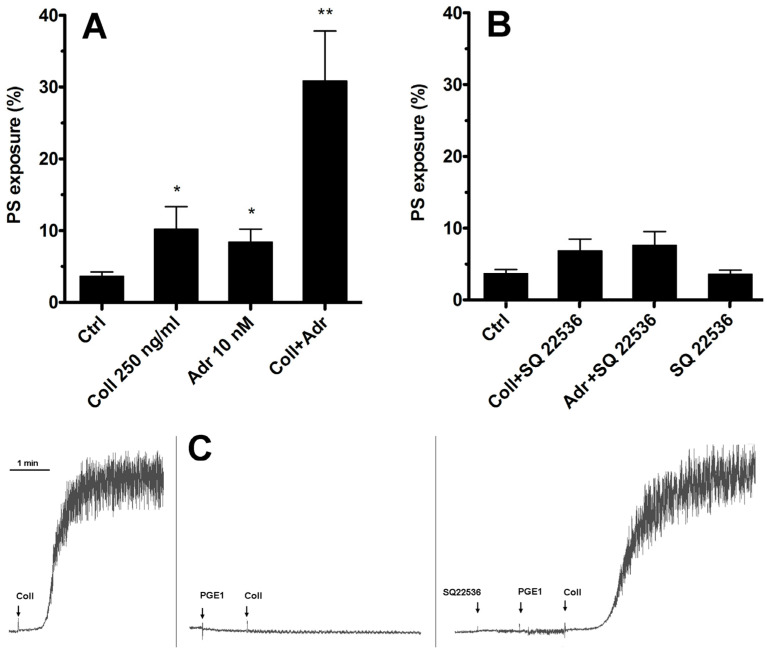
Lack of involvement of adenylate cyclase inhibition in synergistic PS exposure after stimulation of platelet GPVI and α-2-adrenergic receptors. Samples of PRP were incubated with collagen (Coll 250 ng/mL), adrenaline (Adr 10 nM), or with a combination of these agonists (Coll + Adr) (panel (**A**)). Combinations of collagen and SQ 22536 (adenylate cyclase [AC] inhibitor, 50 μM, 10 min of preincubation), adrenaline + SQ 22536, or SQ 22536 alone are presented in panel (**B**). After 15 min of incubation (at room temperature, RT), samples were supplemented with appropriate fluorescent probes, and PS exposure was measured by flow cytometry. SQ 22536 was used at selected concentration (50 μM) that fully recovers aggregability (collagen-induced, 2 μg/mL) of PGE_1_ (potent AC activator, 100 nM)-inhibited platelets. One representative experiment (out of five) is presented (panel (**C**)). Presented are median values with interquartile (1st–3rd) ranges from 5 independent experiments, each in duplicate. Arrows correspond with the moment of indicated compound addition. * *p* < 0.05, ** *p* < 0.01 vs. ctrl.

**Figure 2 ijms-25-02997-f002:**
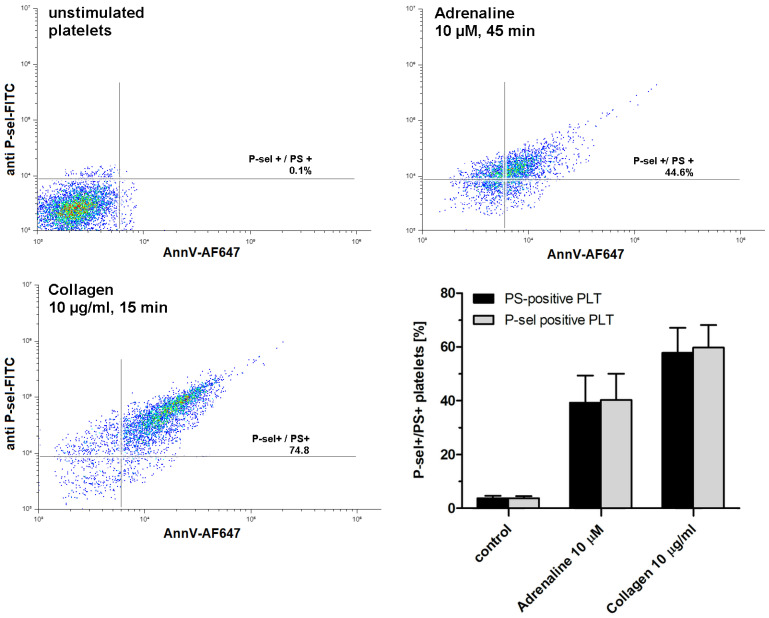
Appearing of procoagulant platelets upon exposure of platelet-rich plasma to adrenaline or collagen. PRP samples were incubated without any additions (control), with adrenaline, or with collagen, added at indicated concentrations. After incubation (at room temperature), samples were supplemented with appropriate fluorescent probes and after final dilution analyzed toward P-selectin appearance and PS exposure by flow cytometry (*n* = 5). Platelets were identified by light scattering properties (forward scatter) and CD41a-related fluorescence (platelet-specific marker). Representative density plots and mean values (±SEM) of the percentage of P-sel+/PS+platelets are presented.

**Figure 3 ijms-25-02997-f003:**
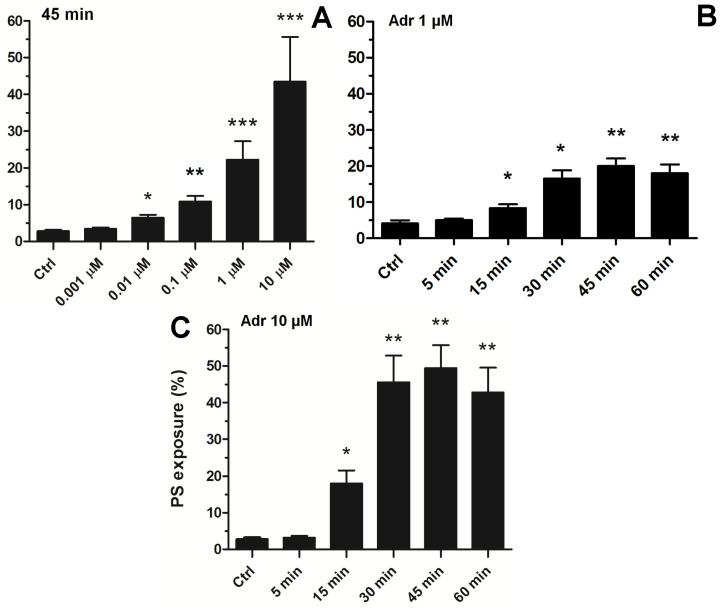
Concentration and time dependence of adrenaline-evoked platelet PS exposure. Samples of PRP were incubated with adrenaline in the following concentrations: 0.001 µM, 0.01 µM, 0.1 µM, 1 µM, 10 µM, or without agonist (Ctrl). After 45 min of incubation (RT), samples were supplemented with appropriate fluorescence probes, and PS exposure was analyzed using flow cytometry (panel (**A**)). The percentage of PS-positive platelets in PRP samples was evaluated by flow cytometry after 5, 15, 30, 45, and 60 min of incubation (RT) without (Ctrl) or with adrenaline at 1 μM (panel (**B**)) or 10 μM (panel (**C**)) final concentration. In both panels, the presented values are medians with interquartile (1st-3rd) ranges from 4 independent experiments, each in three repetitions. * *p* < 0.05, ** *p* < 0.01, *** *p* < 0.001 vs. control.

**Figure 4 ijms-25-02997-f004:**
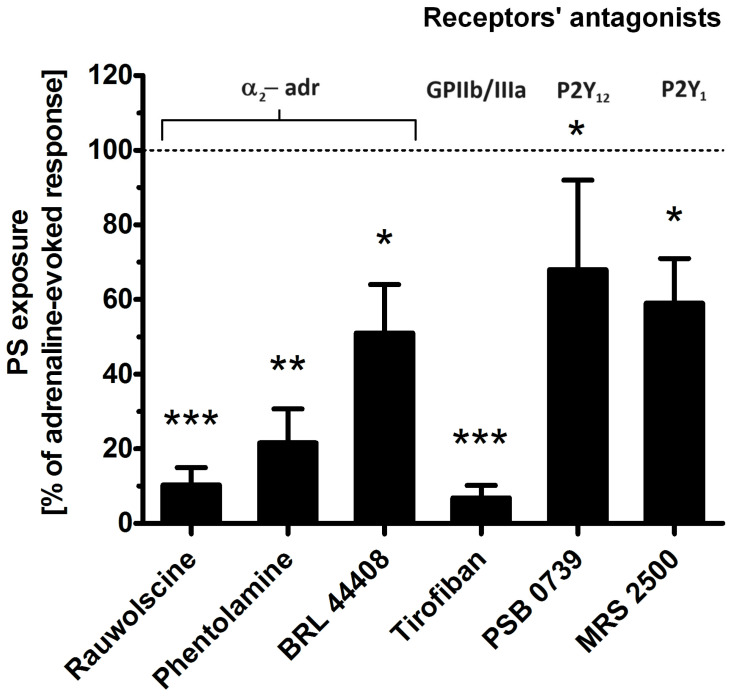
Effect of the antagonists of selected platelet receptors on adrenaline-triggered PS exposure. Appropriate PRP samples after preincubation (10 min) with the following receptors’ antagonists: α_2_-adrenergic (rauwolscine, 10 μM; phentolamine, 10 μM; BRL 44408, 10 μM), GPIIb/IIIa (tirofiban, 500 nM), P2Y_12_ (PSB 0739, 500 μM), and P2Y_1_ (MRS 2500, 1.25 μM) were supplemented with adrenaline (10 μM). In parallel, samples with adrenaline alone (10 μM) and a control sample (without any addition) were prepared. After 45 min of incubation (RT), samples were supplemented with appropriate fluorescence probes, and PS exposure was analyzed by flow cytometry. The values of PS exposure evoked by adrenaline were in the range of 32–55% of PS-exposed platelets out of 10,000 analyzed platelets in each sample. PS exposure evoked by adrenaline (10 μM) was taken as a 100% response for comparative reasons. Presented values are medians with interquartile (1st–3rd) ranges from 10 independent experiments. * *p* < 0.05, ** *p* < 0.01, *** *p* < 0.001 vs. adrenaline.

**Figure 5 ijms-25-02997-f005:**
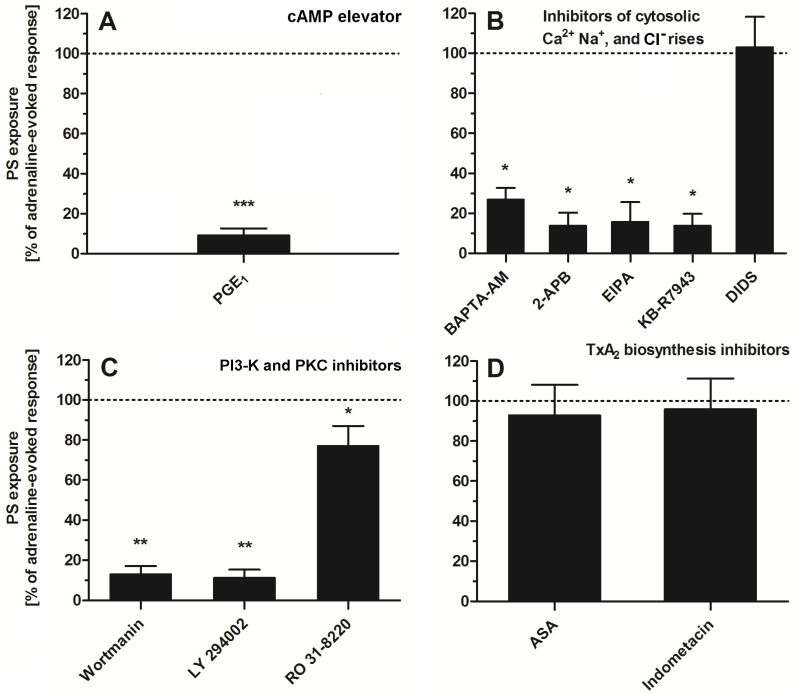
Effect of modulators of selected signaling events on adrenaline-triggered platelet PS exposure. Appropriate PRP samples after preincubation (10 min, except for BAPTA-AM—30 min) with the following modulators of selected signaling events: PGE_1_ (1 µM), (panel (**A**)), BAPTA-AM (62.5 µM), 2-APB (100 µM), EIPA (100 µM), KB-R7943 (50 µM), DIDS (100 µM), (panel (**B**)), wortmannin (200 nM), LY 294002 (100 µM) (panel (**C**)), ASA (200 µM), and indometacine (50 µM) (panel (**D**)) were supplemented with adrenaline (10 μM). Parallel samples with adrenaline alone (10 μM) and a control sample (without any addition) were prepared. After 45 min of incubation (RT), samples were supplemented with appropriate fluorescence probes, and PS exposure was analyzed by flow cytometry. The values of PS exposure evoked by adrenaline were in the range of 32–55% of PS-exposed platelets out of 10,000 analyzed platelets in each sample. PS exposure evoked by adrenaline (10 μM) was taken as a 100% response for comparative reasons. Presented values are medians with interquartile (1st–3rd) ranges from 4–6 independent experiments. * *p* < 0.05, ** *p* < 0.01, *** *p* < 0.001 vs. adrenaline.

**Figure 6 ijms-25-02997-f006:**
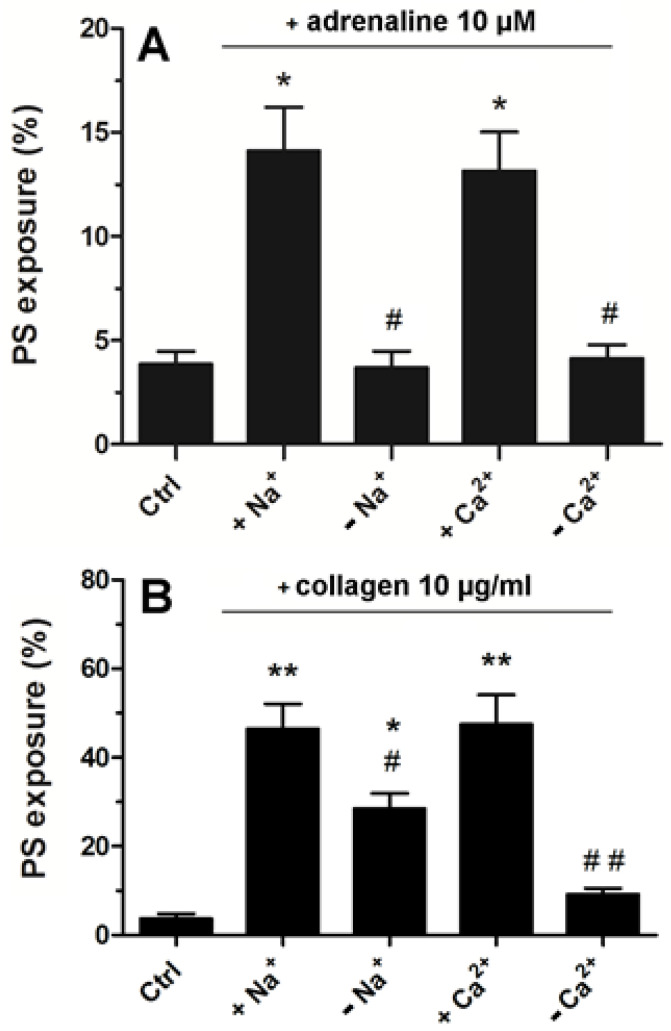
Significance of extracellular Ca^2+^ and Na^+^ for adrenaline- and collagen-evoked platelet PS exposure. Washed platelets were suspended in Tyrode buffer with sodium ions, in Na^+^-free medium, buffer with calcium ions, and Ca^2+^-free medium. Next, the samples were supplemented with adrenaline (10 μM, panel (**A**)) or collagen (10 μg/mL, panel (**B**)). After 45 min of incubation (RT) (15 min for collagen), samples were supplemented with appropriate fluorescence probes, and PS exposure was analyzed by using flow cytometry. The presented values are medians with interquartile (1st–3rd) ranges from 4 independent experiments, each in three repetitions. * *p* < 0.05 vs. control. ** *p* < 0.01 vs control; # *p* < 0.05 and ## *p* < 0.01 vs conditions with Na^+^ or Cl^−^ presence.

**Figure 7 ijms-25-02997-f007:**
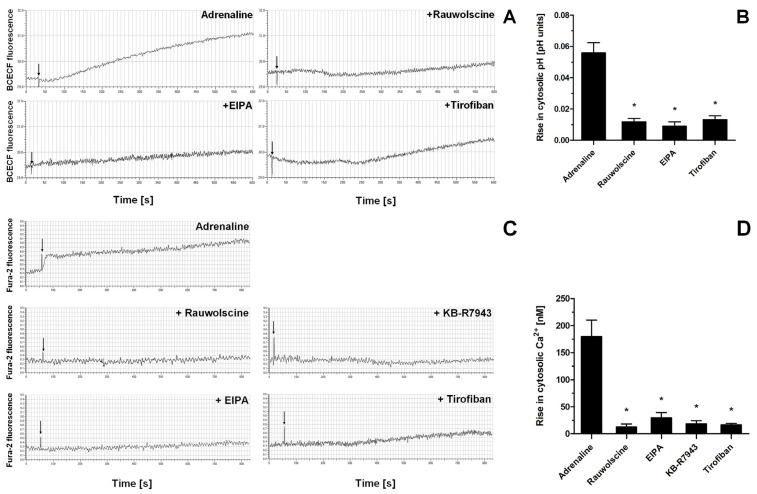
Effect of adrenaline on cytosolic pH value and intraplatelet Ca^2+^ concentration. Samples of BCECF-AM (fluorescent probe sensitive to changes in cytosolic pH)-loaded washed platelets were supplemented with rauwolscine (10 μM), EIPA (50 μM), or tirofiban (500 nM) (preincubation time, 2–4 min) before addition of adrenaline (10 μM). Measurements were conducted in suspension of stirred (800 rpm) and thermostated (to 37 °C) washed platelets. Changes in fluorescence associated with the alterations of platelet cytosolic pH were recorded for up to 10 min (panel (**A**)). Presented values are means ± S.D. from 4 independent experiments. * *p* < 0.05 vs. adrenaline (panel (**B**)). Samples of Fura-2-AM (fluorescent probe sensitive to changes in Ca^2+^ concentration)-loaded washed platelets were supplemented with rauwolscine (10 μM), EIPA (50 μM), KB-R7943 (100 μM), or tirofiban (500 nM) (preincubation time, 2–4 min). Next, measures were initiated by the addition of adrenaline (10 μM). Measurements were conducted in suspension of stirred and thermostated (to 37 °C) washed platelets. Rises in fluorescence associated with Ca^2+^ influx into platelet cytosol were registered for up to 12 min (panel (**C**)). Arrows indicate moment of adrenaline addition into sample.” after the following sentence: Rises in fluorescence associated with Ca^2+^ influx into platelet cytosol were registered for up to 12 min (panel (**C**)). Presented values are medians with interquartile (1st–3rd) ranges from 4 independent experiments. * *p* < 0.05 vs. adrenaline (panel (**D**)).

**Figure 8 ijms-25-02997-f008:**
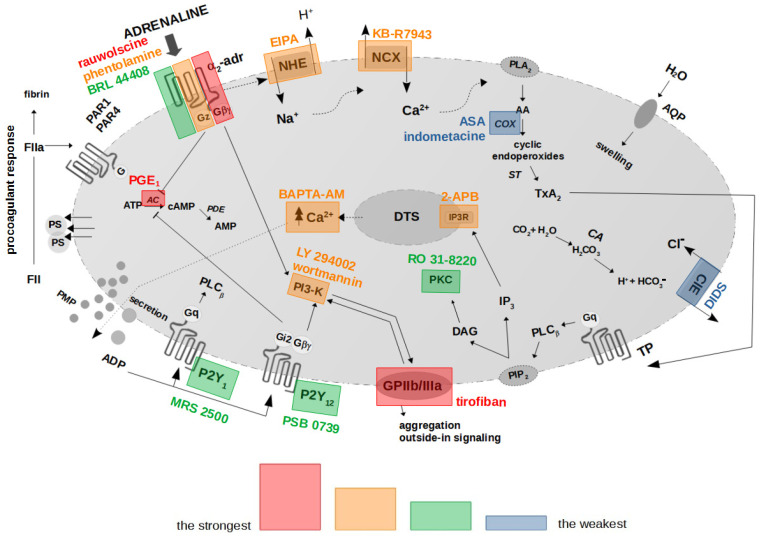
Proposed functional model of platelet activation by adrenaline: effect of antagonists of selected platelet receptors and modulators of selected signaling events on adrenaline-triggered platelet PS exposure. The colored areas indicate the strength of procoagulant response reduction by modulators of signal pathways/platelet receptors’ antagonists. α_2_-adr—α_2_-adrenergic receptor; AA—arachidonic acid; AC—adenylyl cyclase; ADP—adenosine diphosphate; ATP—adenosine triphosphate; AMP—adenosine monophosphate; cAMP—cyclic adenosine monophosphate; ClE—Cl^−^/HCO_3_^−^ exchanger; COX—cyclooxygenase; DAG—diacylglycerol; DTS—dense tubular system; FII/FIIa—prothrombin/thrombin; IP_3_—inositol 1,4,5-trisphosphate; IP3R—inositol 1,4,5-trisphosphate receptor; NCX—Na^+^/Ca^2+^ exchanger; NHE—Na^+^/H^+^ exchanger; P2Y_1_—ADP receptor; P2Y_12_—ADP receptor; PAR1—protease-activated receptor-1; PAR4—protease-activated receptor-4; PI3-K—phosphoinositide 3-kinase; PIP_2_—phosphatidylinositol 4,5-bisphosphate; PKC—protein kinase C; PLA_2_—phospholipase A_2_; PLC_β_—phospholipase C_β_; PMP—platelet microparticles; PS—phosphatidylserine; ST—thromboxane synthase; TP—thromboxane A_2_ receptor; TxA_2_—thromboxane A_2_. Blunt arrows (┴) indicate inhibition while sharp arrows (→) indicate stimulation.

## Data Availability

The data presented in this study are available upon request from the corresponding author.

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
