# Peer review of "Study on the Mechanism of the Adrenaline-Evoked Procoagulant Response in Human Platelets"

_ijms, 2024, doi:10.3390/ijms25052997_

Round 1

Reviewer 1 Report

Comments and Suggestions for Authors

The study by Golaszewska et al. addresses an important question of how adrenalin enhances the progoagulant activity of platelets in a prothrombotic environment. The question is clear and the flow-cytometry technique they used is very relevant. However, there are some missing controls and experiments required to answer their question. To improve their study design, the authors have to investigate the effect of all the inhibitors and antagonists on collagen/thrombin/ADP and adrenalin-induced PS exposure. Their results on adrenalin alone-induced PS exposure is not sufficient.

Other comments:

1. Figure 1- What is the proof that SQ22536 treatment actually worked in this experiment? The authors should include some evidence to show that SQ22536 is functional.

2. Figure 1- SQ22536-alone treated control is missing. It will be easy to include it in the graph rather than mentioning ‘data not shown’ in the legend.

3. Figure 3- The authors failed to explain why 3 different α2-adr antagonists produce very different values in PS exposure?

4. Figure 3 and 4: It is important to repeat these experiments with adr and collagen-induced platelets.

5. Figure 5: I don’t think this data is important here as we know the role of Ca2+ from text books.

Reviewer 2 Report

Comments and Suggestions for Authors

This manuscript from GoÅ‚aszewska et al is a continuation of the published in 2021 paper where they described that adrenalin induces procoagulant platelet formation. In the presented manuscript, they analyzed intracellular molecular mechanisms of platelet PS surface exposure and came to the conclusion that outside-in signaling of GPIIb/IIIa is the main trigger of α2-AR receptor mediated PS surface exposure. All experiments are well performed and the manuscript is clear written, however there are still some questions that the authors should take into account.

1.      Results. 2.1. Something is not logical here. In this part should be shown the results of adrenaline + AC inhibitor, but not collagen + AC inhibitor. Also will be good to show different adrenaline concentrations with AC inhibitor. SQ 22536 is a well known AC inhibitor, however it is not always working consistently and it should be controlled by measurement of cAMP, or PKA activity. Collagen does not increase platelet cAMP level, therefore it is not clear why they used SQ + collagen instead of SQ + adrenaline.

2.      Platelet surface PS exposure may be mediated by two independent mechanisms (procoagulant platelet formation, or by apoptotic pathway). The authors should show that extremely high (10 µM) adrenalin concentration did not activate apoptotic pathways in platelets.

3.      Results 2.2. It is not clear why the authors decided to use such high (10 µM) concentration for all future experiments. In the part “How does this study adhere to clinical reality?” they mentioned that the maximum adrenalin concentration after infusion is in nM range. Probably 1 µM and incubation 30 min will be better and more relevant for this study. At least will be important to show this result (1 µM, 30 min).

4.      Results 2.3. According to these data all the effects of adrenalin on PS exposure is mediated only by GPIIb/IIIa outside-in signaling. This looks very strange and  no real explanation in the discussion.

5.      Results 2. 4. Acetazolamide is the inhibitor of carbonic anhydrase and HAuCl4 might have unspecific inhibitory, not related to adrenaline, effects on platelets. This should be tested separately. Inhibition of carbonic anhydrase will itself inhibit platelet activation, not related to adrenaline. For analysis of cyclic nucleotide effects will be better to use specific inhibitor of PDE5 (sildenafil) and PDE2, or IBMX, but not PDE3 which plays a minor role in among all known PDEs expressed in platelets (PDE5, PDE2, PDE3). In any case cAMP, cGMP concentrations should be measured in these experiments.  

6.      Results 2.5. How specific are the experiments without Na for adrenaline. Some controls should be presented here. For example similar experiments with collagen and thrombin. The same for calcium experiments. Without such controls all explanations that it is connected to adrenaline effects are not correct.

7.      Discussion 3. 1. The conclusion of this part has no real basis (see also point 1). For such conclusion the authors should do experiments with different doses of adrenaline + SQ and control SQ effects by measuring cAMP concentration or PKA activity.

8.      3. 2. “Such high (supraphysiological) concentration of adrenaline was able to evoke strong PS exposure, comparable to the effect exerted by collagen, the most procoagulant one between physiological platelet agonists.” In the Figure 1. Collagen only moderately (ca. 2 times) increased PS exposure. In the Fig. 2 adrenalin induces much higher PS exposure.

9.      3.7. How does this study adhere to clinical reality?  Seems that such extremely high adrenalin concentrations have no connections with clinical reality. Even the authors on the end of this section mentioned that “Alternative candidate to restore platelet aggregability in ticagrelor-receiving patients could be bentracimab.”

Reviewer 3 Report

Comments and Suggestions for Authors

The study focusses on the role of adrenaline in mediating platelet PS exposure. 

Major critique:

- What type of PS+ platelet is this, are these apoptotic platelets, or procoagulant (necrotic) platelets or a combination of both. 

The readouts used in this manuscript are inappropriate and no longer considered standard in the field. Please refer to the following manuscript for appropriate assessment of the type of PS+ platelet. (PMID: 37172731)

The authors need to demonstrate more markers that can distinguish between both. 

Other critiques:

- The authors need to validate the AC inhibitor and show it effectively blocks it in their hands/experimental setup. 

- How relevant are the concentration of adrenaline used? Why not combine it with a GPVI agonist as previously. 

- The time it takes for the PS to rise on the platelet surface doesn't seem to indicate an important role for adrenaline in this process (at least alone). Usually PS+ platelets are formed within a couple of minutes of stimulation. 

- The studies using the inhibitors don't make sense and completely go against all literature studying PS+ platelets. How would tirofiban prevent this proces for example. The authors need to validate these findings with different doses of these drugs and additional drugs with similar modes of action. 

- How specific is the effect of the ions tested to adrenaline, isn't this true for all platelet assays?

Round 2

Reviewer 1 Report

Comments and Suggestions for Authors

The manuscript can be accepted in the current format.

Reviewer 2 Report

Comments and Suggestions for Authors

The authors adequately addressed all critics and I have no more questions

Reviewer 3 Report

Comments and Suggestions for Authors

The authors addressed most of the reviewers critiques.